# The Candoglia Marble and the "Veneranda Fabbrica del Duomo di Milano": A Renowned Georesource to Be Potentially Designed as Global Heritage Stone

**Giovanna Antonella Dino [1],\*, Alessandro Borghi [1], Daniele Castelli [1], Francesco Canali [2], Elio Corbetta [2] and Barry Cooper [3]**

1   Department of Earth Sciences, University of Torino, Via Valperga Caluso, 35, 10125 Torino, Italy
2   Veneranda Fabbrica del Duomo di Milano, Via Carlo Maria Martini, 1, 20121 Milano, Italy
3   School of Natural and Built Environments, University of South Australia, Adelaide SA 5001, Australia
\*   Correspondence: giovanna.dino@unito.it; Tel.: +39-011-670-5150

**Abstract:** Marbles from Alpine area have been widely employed to build and decorate masterpieces and buildings which often represent the cultural heritage of an area (statuary, historic buildings and sculptures). Candoglia marble, object of the present research, is one of the most famous and appreciated marbles from Alpine area; it has been quarried since Roman times in the Verbano-Cusio-Ossola (VCO; Piemonte—NW Italy) extractive area. Candoglia Marble outcrops are present as lenses within the high-grade paragneisses of the Ivrea Zone, a visible section of deep continental crust characterised by amphibolite- to granulite-facies metamorphism (Palaeozoic period). Candoglia calcitic marble (80–85% $CaCO_3$ and the 15–20% other minerals) shows a characteristic pink to gray colour and a coarse-grained texture (>3 mm): frequent centimetre-thick dark-greenish silicate layers (mainly represented by diopside and tremolite) characterize the texture of the marble. It has been largely used in local rural constructions and historical buildings, but its most famous application has been (and still is) for the "Duomo di Milano" construction (fourteenth century). The Veneranda Fabbrica del Duomo di Milano carried out the anthropogenic activities dealing with the Candoglia marble exploitation; it has to be highlighted that the company have managed the Marble exploitation during the last seven centuries and that the quarry itself is a tangible sign of the development of extraction and heritage in the VCO area. Candoglia marble can be recognized as a significant example of a "Global Heritage Stone Resource": its exploitation from quarry to building (the Duomo di Milano) well represents the close correlation between stone and cultural heritage, between georesources and humankind development.

**Keywords:** Candoglia marble; Duomo di Milano; petrographic analysis; geoheritage; cultural stone

## 1. Introduction

Italy has been the cradle of several cultures and architectonic styles, from the Roman period to the present. This has left tangible traces, such as historical buildings, sculptures, etc., many of which are made of a multitude of ornamental stones, frequently sourced from the quarries present in the area. This cultural heritage represents the tangible signs of the historical and cultural wealth of an era in specific territories. A clear example of this "stone culture" is represented by the wide utilization of white marble (e.g., the Luni marble, now recognized as Carrara marble) in both *statuaria* and architecture since the Greek and Roman times [1].

Marble varieties from the Italian Western Alps, even though they are not as famous as the renowned Carrara varieties, have been largely employed for sculptures and buildings construction.

The Piemonte region has been and still is strongly interested by intensive quarrying activity: exploited stones were and are largely employed in cultural heritage, from the Roman times (e.i. Arc of Augustus in Susa (9th BC.) [2]) to the late Baroque period (e.i. Savoy architecture in Turin [3]). Nevertheless, few publications report on the petrographic and geochemical characteristics from an archaeometric point of view; moreover, often, the Piemonte varieties are not included in the provenance databases of classical marble [4–7].

Generally, the stones used for ornamental purposes are documented only via historical documents or macroscopic data [8]. For a few cases, for example for the Arc of Augustus or the churches of St. Cristina and St. Filippo Neri in Turin, a deeper scientific analysis was carried out on marble from the Western Alps for stone provenance definition [9,10]. More recently, a minero-petrographic and isotopic overview of the main Piemonte marble varieties has been reported [11]. Thus, knowledge on stone resources (mineral-petrographic characteristics, past and present quarrying and working techniques, applications in local, national and international buildings and/or architectures, etc.) is fundamental to emphasize the historical and cultural relevance of such georesources, highlighting the importance of an economic activity, which is fundamental for the cultures and customs of the different heritages that have characterized and strengthened the Mediterranean area through the centuries.

It is fundamental to share knowledge on these historical and cultural stones among a wider audience rather than storing this knowledge and disseminating it only among researchers and experts, so as not only to improve general knowledge on natural resources but also to enhance the self-consciousness of the deep connection between environment and exploitation, balancing both sustainability and cultural heritage. The aim of this paper is to disseminate knowledge on Candoglia marble, a pink, coarse-grained marble, locally known from the late Middle Ages (with renowned applications for the Madonna di Campagna Church at Verbania and the San Giovanni Battista Church at Montorfano, VCO Province, Northern Piemonte), and made famous thanks to its use in the Veneranda Fabrica del Duomo di Milano in the fourteenth century. It was also used for the facades of different monuments such as the Certosa di Pavia (Carthusian monastery), the Cappella Colleoni (Bergamo) and the San Petronio Cathedral (Bologna); moreover, in the past, it was used for altars and columns in Roman buildings (e.g., two columns in the Chiostro Arcivescovile at Novara, Piemonte) [12].

The study of the investigated material was carried out using different analytical techniques, both traditional (such as optical microscopy and stable isotope analysis) and more modern (Scanning Electron Microscope (SEM), equipped with an Energy Dispersion Spectrometry (EDS) analytical facilities) such approach rarely has been applied, to date, on Candoglia marble.

Furthermore, the development of quarrying and working techniques, together with some information about the Veneranda Fabbrica del Duomo di Milano, are presented in this paper.

## 2. Geological Setting

Palaeozoic marbles include the Ornavasso and Candoglia ones which pertain to the Verbano–Cusio Ossola (VCO) quarrying basin (Southalpine geologic domain).

In Piedmont territory, Southalpine domain is represented by deep and intermediate continental crust units, denominated Ivrea-Verbano zone and "Serie dei Laghi". They are separated by tectonic lineaments of Cossato-Mergozzo-Brissago. Ivrea-Verbano zone is mainly composed of Permian age igneous bodies ([13], with refs.). One of the igneous body is represented by the renowned Ivrea gabbro batholith, which was put in place at the base of a thinner gneissic crust (Kinzigitic complex; [14]). The "Serie dei Laghi", which crops out in S-E Ivrea-Verbano, consists of upper-intermediate tectonic unit which suffered an earlier metamorphic event in the Ordovician age.

Lenses of marble (up to 30m thick) are inserted within the amphibolite-granulite paragneisses characterising the Ivrea Zone. The Ornavasso and Candoglia marbles are quarried in the two described areas in which the lens of marble outcrops (on the right and left sides respectively of the Toce River–Ossola Valley), few kilometres west of the mouth of the river in the Lake Maggiore (Figure 1b).

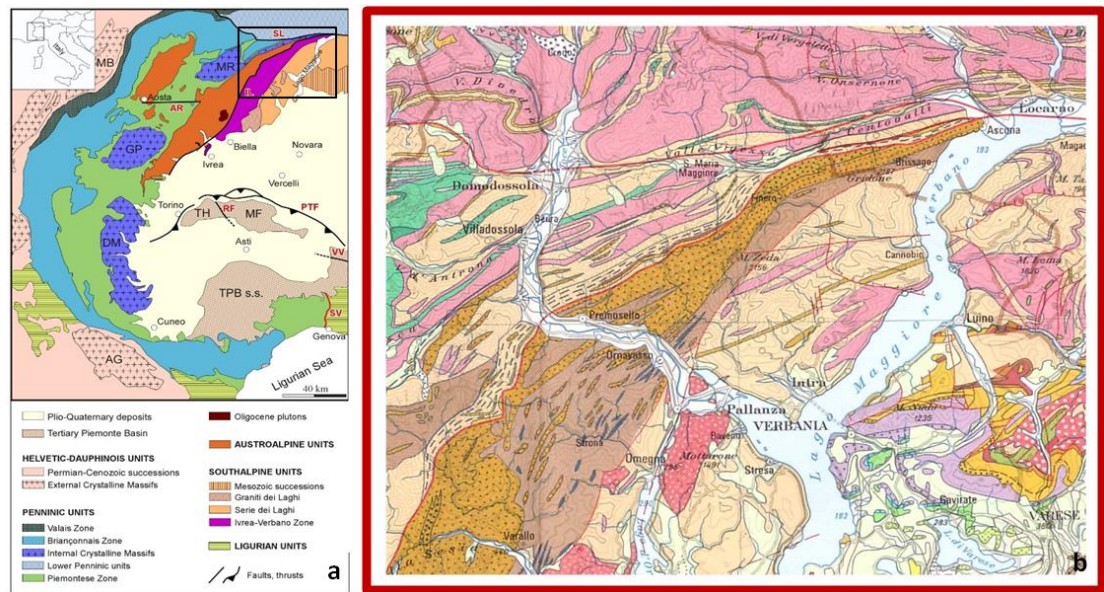

**Figure 1.** (**a**) Tectonic sketch-map of the Western Alps. Tectonic lines: IL = Insubric Line; SL = Simplon Line; SV = Sestri Voltaggio Line, RF = Rio Freddo Line, VV = Villalvernia-Varzi Line; AR: Aosta Ranzola Fault; PTF: Apennine Front Thrust. MB = Monte Bianco; MR = Monte Rosa; GP = Gran Paradiso; DM = Dora Maira; AG = Argentera; TH = Torino Hill; MF = Monferrato; TPB = Tertiary Piedmont Basin. The square correspond to the geologic map reported in Figure 1b. (**b**) Geological map of the lower Ossola Valley (from Domodossola to Verbania, on the western branch of Lago Maggiore) [15]. Marble of the Ivrea Zone are reported as North-North East (NNE) South-South West (SSW) trending, thin lenses (given in blue colour) within the high-grade paragneiss (given in brown).

## 3. Candoglia Marble Characterisation: Materials and Methods

### 3.1. Petrographic and Geochemical Characterisation

Petrographic analysis by optical microscope and SEM, together with the minero-chemical analysis of the main and accessory minerals by EDS microanalysis and the micro X-ray florescence determination of trace elements were used to characterise six samples representative of two historical quarries (Table 1).

**Table 1.** Candoglia marble samples localisation.

| Sample | Quarry Name |
|--------|-------------|
| canAB1 | Cava Madre underground pit |
| canAB2 | Cava Madre underground pit |
| canAB3 | Cava Madre underground pit |
| canAB4 | Cava Corte Nuova open pit |
| canAB5 | Cava Corte Nuova open pit |
| canAB6 | Cava Corte Nuova oper pit |

Petrographic analyses were carried out on polished thin sections, using a Cambridge S360 scanning electron microscope, according to the analytical conditions reported in [16]. In particular, the working distance was 25 mm; the probe current was 200 pA; the accelerating potential was 15 kV; the counting time resulted of 60 s. All the analyses were recalculated with MINSORT program by Petrakakis and Dietrich [17].

Micro-XRF Eagle III-XPL (Röntgen analytic Messtechnik GmbH, Butzbach, Germany) was used for trace elements analysis of calcite and dolomite in rock samples. Analytical condition are reported in [18]. The Candoglia marble archaeometric characterization has been carried out using stable isotopic data reported by Antonelli and Lazzarini [19].

### 3.2. Physico-Mechanical Characterisation

To characterise a stone, it is fundamental, together with the minero-petrographic characterisation, to know its physico-mechanical behaviour. Data about physico-mechanical characterization (such as apparent density, water absorption, uniaxial compressive strength, compressive strength after freeze and thaw, indirect tensile strength and impact strength test) have been collected from published papers and official web sites. The legislation followed to carry out the different tests is reported in the consulted publications [20–22].

## 4. Candoglia Marble Characterization: Results

### 4.1. Petrographic and Geochemical Characterisation

Candoglia marble shows a characteristic pink to gray colour and a coarse-grained texture (>3 mm); furthermore it is characterised by the 80–85% $CaCO_3$ and the 15–20% other minerals, concentrated along discrete layers. The greyish aspect is due to diopside and tremolite (centimetre-thick) (Figure 2a); minor minerals include epidote, quartz, barite, Ba-feldspar, sulphides and, rarely, phlogopite. It shows a heteroblastic texture with a sutured grain boundary shape of the carbonate crystals (Figure 2b). The structure is isotropic as no preferred crystallographic orientation of calcite grains has been evidenced. The average grain size is 0.28 ± 0.1 mm and the high standard deviation is linked to the heteroblastic microstructure, which results in a maximum grain size (MGS) showing a median value of 3.3 mm (from 2.6 to 4.9 mm). Such a coarse grain characteristic, exceeded by varieties of Mediterranean marble only from Naxos marble and Brossasco marble [11,19], did not prevent this marble from being widely used in architecture and statuary. The marbles coming from VCO quarrying basin are also characterized by values of oxygen more negative than other marble of the Western Alps. For Candoglia marble, a range between −7 and −13 $^{18}\delta O$ is reported by [19]. This feature is probably due to the high metamorphic grade reached by the Ivrea Zone, the geological unit to which Candoglia marble pertains.

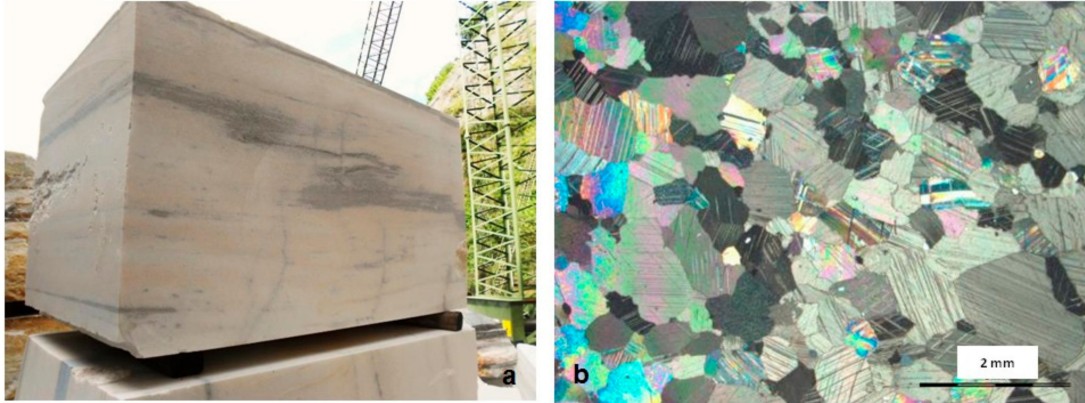

**Figure 2.** (**a**) Macroscopic appearance of marble: the presence of dark-greenish silicate layers (centimetre-thick) which highlight the suffered ductile deformation is clearly visible. (**b**) Photomicrograph of marble (can AB2), showing the heteroblastic texture, with the sutured grain boundary shape of the carbonatic crystals. The structure is isotropic as no preferred crystallographic orientation of calcite grains has been evidenced. Optical microscope crossed nicols.

Candoglia marble also shows distinctive minor and trace element contents, which help in characterizing and distinguishing it from all the other Mediterranean marbles. The manganese and iron contents are respectively 119 ± 3 and 186 ± 4 ppm, reflecting the chromatic features of the marble. Despite remarkable macroscopic heterogeneities, the Fe and Mn concentrations do not show significant variations within all the analysed areas. The Sr concentration results in a mean of 212 ± 30 ppm, which is significantly higher than most of the investigated Mediterranean varieties. Quantitative results, referring to micro-X-Ray Fluorescence (XRF) calcite trace element concentration, are reported in Table 2.

**Table 2.** Geochemical and minero-petrographic variables characterizing Candoglia marble reference samples. Micro-XRF trace element composition of calcite has been recalculated based on 43.97% $CO_2$. AGS and MGS refer to average and maximum grain size, respectively, while the microstructural and textural features are expressed as categorical variables. Ho/He refers to homeoblastic vs. heteroblastic microstructure; Iso/Aniso to isotropic vs. anisotropic microstructure. The grain boundary shape (GBS) is, in this, case sutured (Sut).

| Sample | Mg (%) | Ti (ppm) | Mn (ppm) | Fe (ppm) | Zn (ppm) | Sr (ppm) | AGS | MGS | Ho/He | Fabric | GBS |
|---|---|---|---|---|---|---|---|---|---|---|---|
| **canAB1** | 0.418 | 2 | 119 | 179 | 4 | 228 | 0.29 | 3.51 | He/Ho | Iso | Sut |
| **canAB2** | 0.296 | 2 | 119 | 189 | 5 | 216 | 0.32 | 3.12 | He/Ho | Iso | Sut |
| **canAB3** | 0.378 | 2 | 117 | 189 | 7 | 241 | 0.20 | 3.09 | He/Ho | Iso | Sut |
| **canAB4** | 0.393 | 2 | 116 | 187 | 3 | 186 | 0.39 | 3.39 | He/Ho | Iso | Sut |
| **canAB5** | 0.417 | 2 | 120 | 187 | 2 | 202 | 0.46 | 3.47 | He/Ho | Iso | Sut |
| **canAB6** | 0.489 | 2 | 122 | 187 | 2 | 202 | 0.34 | 3.16 | He/Ho | Iso | Sut |

The peculiar pink colour, the coarse grain size and the isotopic value of 18δO are all archaeometric characteristics that make it easy to identify this marble from any other Mediterranean marble used in local and international cultural heritage.

*4.2. Physico-Mechanical Characterization*

Table 3 reports the more common physico-mechanical characteristics tested for Candoglia marble, compared to the famous Carrara white marble. The characteristics of both are comparable except for three values: the water absorption (higher in Candoglia marble), compressive strength after freeze and thaw (lower in Candoglia marble) and impact strength (lower in Candoglia marble). The first two characteristics are linked to the feature of the rocks: coarse grain size in Candoglia marble and finer grain size in Carrara marble.

**Table 3.** Physico-mechanical characterisation of Candoglia marble.

| Characteristics | Unit | Candoglia Marble | Carrara Marble *** |
|---|---|---|---|
| Apparent density | kg/m$^3$ | 2620–2830 * | 2688 |
| Water absorption | % | 0.8 ** | 0.16 |
| Uniaxial compressive strength | MPa | 44–155 * | 118.6 |
| Compressive strength after freeze and thaw | MPa | 95 ** | 115.8 |
| Indirect tensile strength | MPa | 1.8–12.3 * | 17 |
| Impact strength test | cm | 38 ** | 73.8 |

* data from [20]; ** data from [21]; *** data from [22].

## 5. Candoglia Marble Exploitation and Processing

The first traces of the exploitation of marble outcrops in the Ornavasso territory belongs to the Roman period; the marble presents in Ornavasso pertains to the same marble lenses of Candoglia marble (Figure 1) and is locally recognized as the Candoglia "bastard brother". Both the Candoglia and Ornavasso marbles have been widely employed in local buildings and infrastructures; the more noble variety, Candoglia marble, is renowned for its applications in the Milan Cathedral: in the 1387 Gian Galeazzo Visconti obtained the authorisation to exploit the Candoglia Marble. From that time, Candoglia marble quarries have been quarried by the Veneranda Fabbrica del Duomo of Milano and the exploited material, nearly 1.000 t/year, has been and still is employed uniquely for building and maintenance of the Cathedral [23,24].

The quarrying activity evolved over the centuries: from small open pits, near Candoglia village, to the large underground quarry (Cava Madre) (Figure 3; Figure 4). Cava Madre is characterized by homogeneous materials, exploited in subvertical benches (Figure 5), first using sledgehammers, *punciotti* (plug and feather), chisels and gunpowder (Figure 6a) and later using heilcoidal wires (Figure 6b) and

diamond wires [12]. The bench is interested by side forces, that need to be constantly monitored and contained [20,25–27]. The activities in the quarry yard were characterized by initial selection of the best blocks, chosen on the basis of the aesthetical, physical, and lithological characteristics.

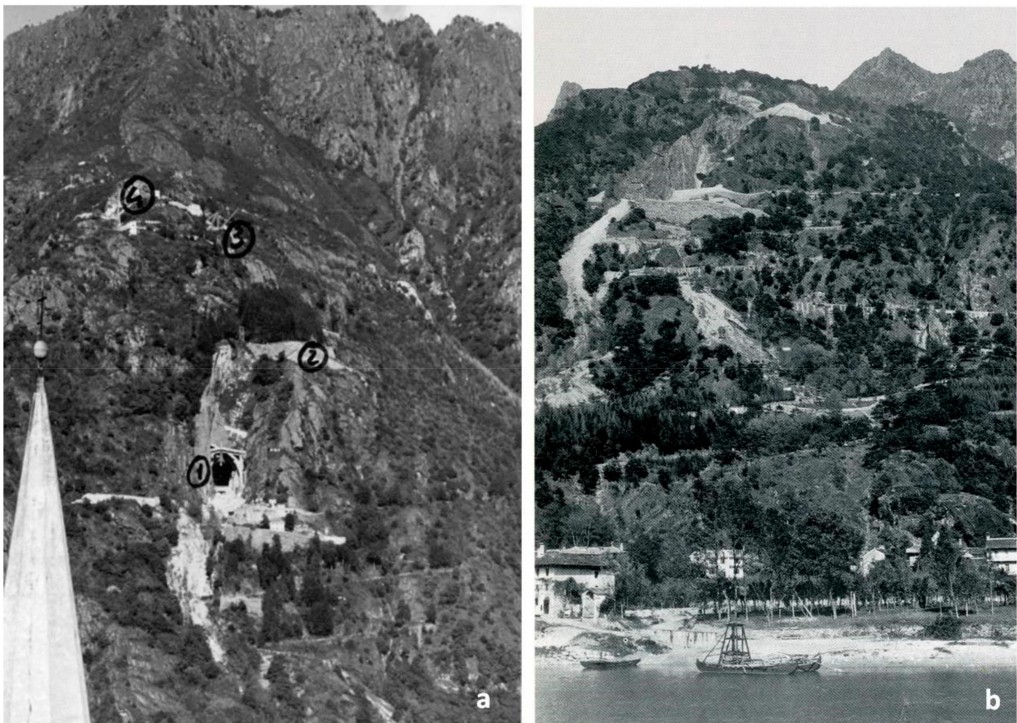

**Figure 3.** (**a**) View (and sketch) of the Candoglia quarries (from San Nicola church in Ornavasso): 1. Cava Madre underground pit; 2–4. Cava Corte Nuova open pits, active up to the early XX cent. (**b**) View of the Candoglia open pit quarries and harbour (from Ornavasso side). Late XIX cent. (Veneranda Fabrica Historical Archive).

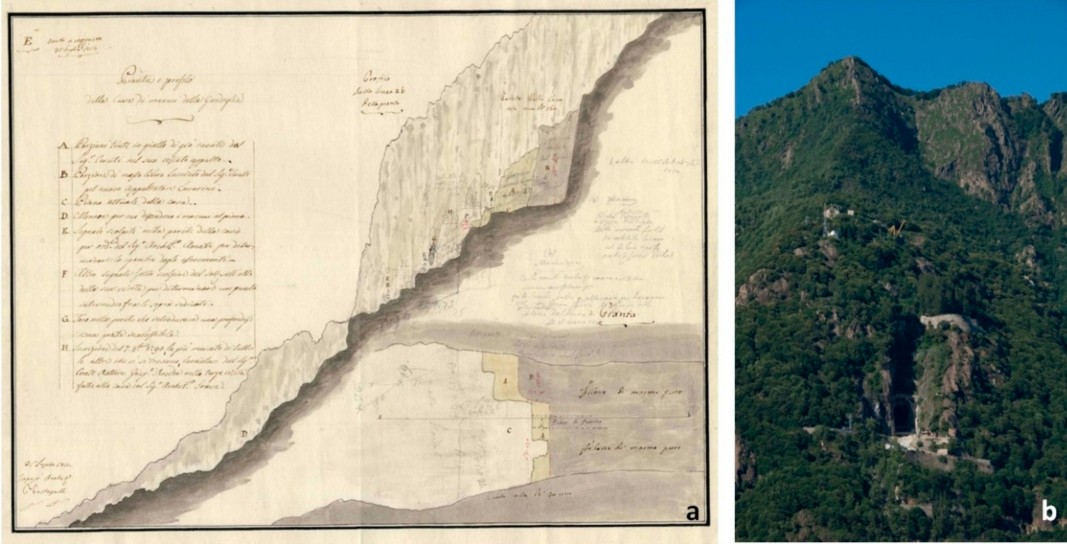

**Figure 4.** (**a**) Cava Madre quarry profile (1814) (Veneranda Fabrica Historical Archive). (**b**) Cava Madre entrance: view from San Nicola Church bell tower (Ornavasso).

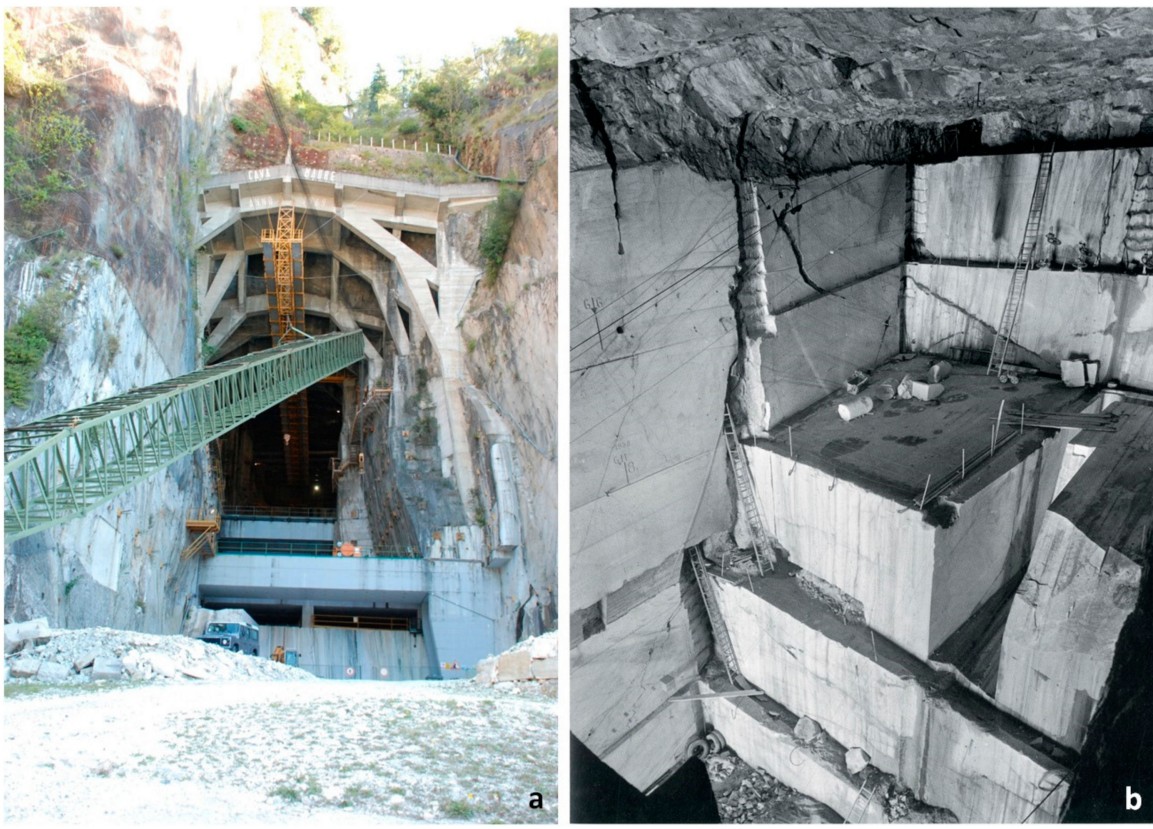

**Figure 5.** (**a**) Entrance of Cava Madre underground quarry. (**b**) Cava Madre underground pit. XX century. (Veneranda Fabrica Historical Archive).

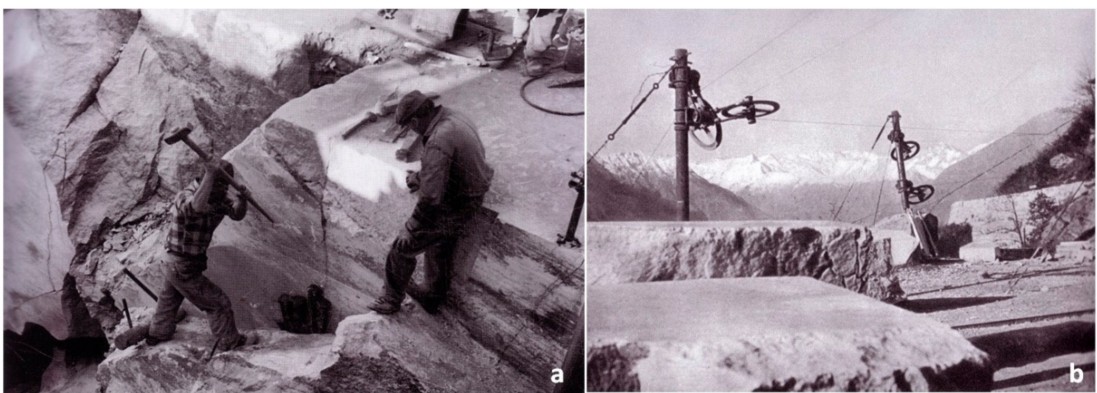

**Figure 6.** (**a**) Quarry worker with a sledgehammer, *punciotti* (plug and feather), and chisels. (**b**) Candoglia open pit yard: helicoidal wire. (Veneranda Fabrica Historical Archive).

The selected blocks were squared directly in the quarry yard: a first rough sketch of the sculpture to be produced was arranged in the yard, based on sketch drawings of blocks and sculptures (Figure 7), before being transported to the working plants to be chiselled and refined. The less valuable blocks (not aesthetically good) were sold as a secondary category material, which was crushed and used to produce lime (traces of kilns are still visible in region Calmatta, near Ornavasso Village) [28].

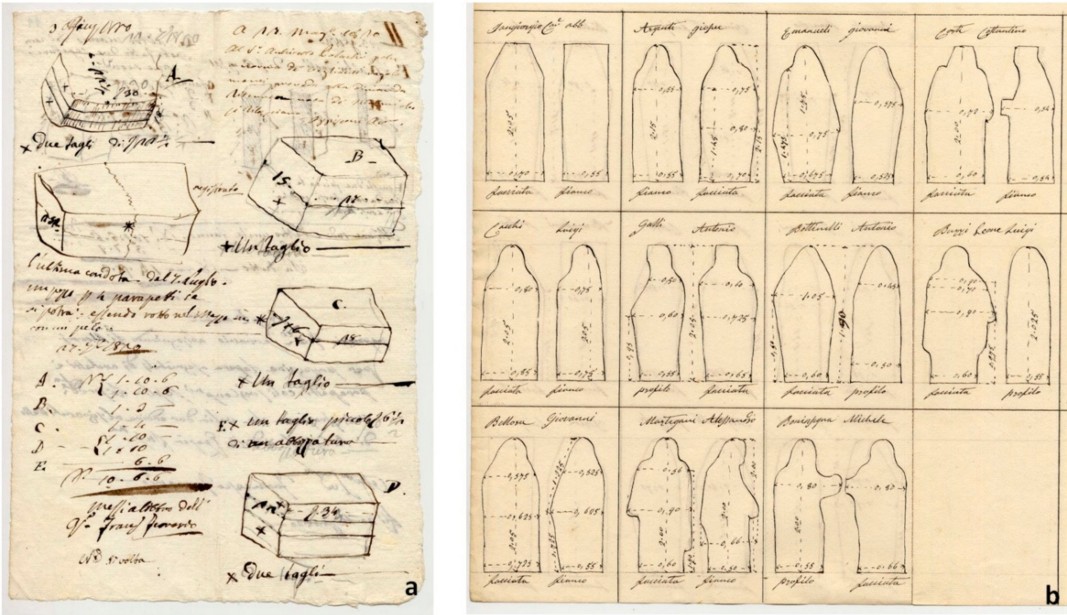

**Figure 7.** (**a**) Sketch drawings of blocks, with defects. XVIII Cent. (**b**) Sketch of sculptures, each assigned to a single stone worker. (Veneranda Fabrica Historical Archive).

After being quarried and squared, the blocks were transported from the quarry to the manufacturing plants on running routes, thanks to the employment of large wooden sledges—"struse" (Figure 8). More recently (from the 1920), the "binda" (jack) or the derrick were used to move the blocks. The blocks were then transported to the river docks by means of wagons towed by pairs of oxen (Figure 9a). From the River Toce docks, the marble was transported to small manufacturing plants (Figure 9b), in Suna and Baveno (along Lake Maggiore); in these laboratories, all the locally exploited blocks were squared, cut and worked. It has to be highlighted that the first mechanized saw for blocks was built for the processing of Candoglia marble.

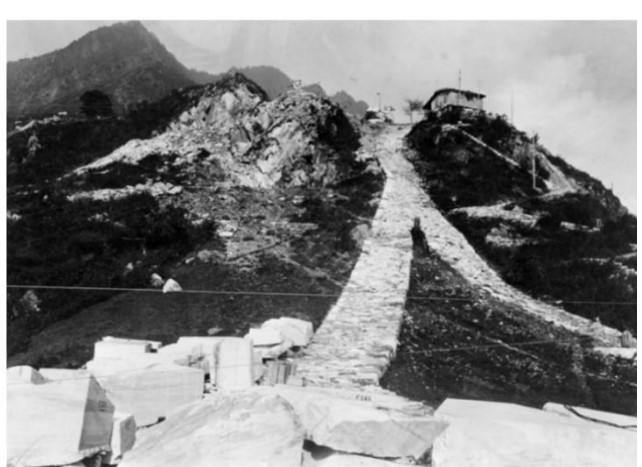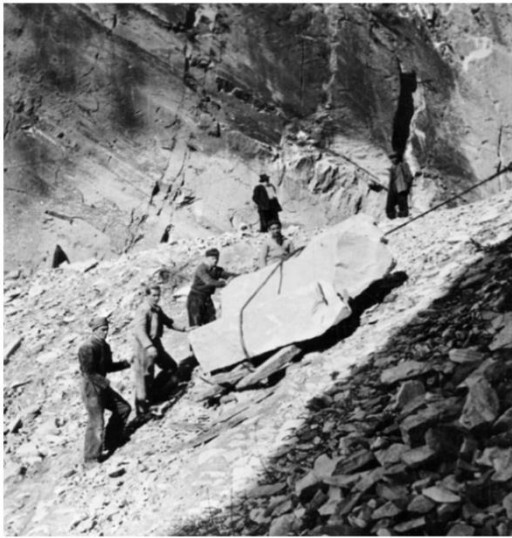

**Figure 8.** Running routes made of quarry waste (left) and big wooden sledges—"struse"—to transport the blocks from the quarry to the valley (right) [29].

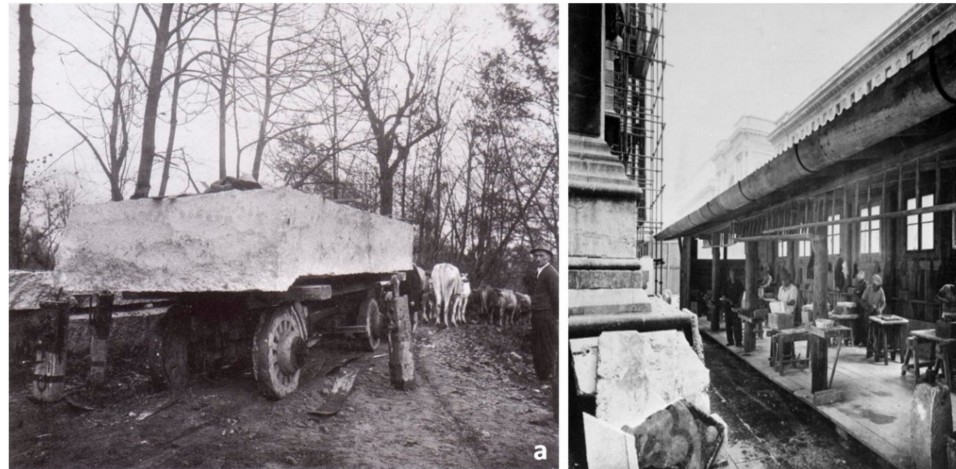

**Figure 9.** (**a**) Wagons towed by pairs of oxen which transported the blocks from the valley to the river pier. (**b**) Veneranda Fabbrica working yard in Milan (XIX–XX cent.) [29].

After the working phase, the blocks were transported, using big barges, from the Candoglia pier (Figure 10a) to the Navigli in Milan (Figure 10b), up to the Duomo di Milano working yard: the path was through the River Toce, the Lake Maggiore and the River Ticino. The presence of rivers (Toce and Ticino) and the construction of canals (Navigli) guaranteed the direct and easy transport from the quarry area to the Cathedral yard. Thus, Candoglia marble was preferred to other more famous Italian marble (e.g. Carrara marble) even if its minero-petrographic features (coarse grain-size) were not the most suitable to use for statuary applications. It has to be highlighted that until mid-XV century, approximately 400 people were employed as stone workers for the Duomo construction, with another 400 people as transport workers [29].

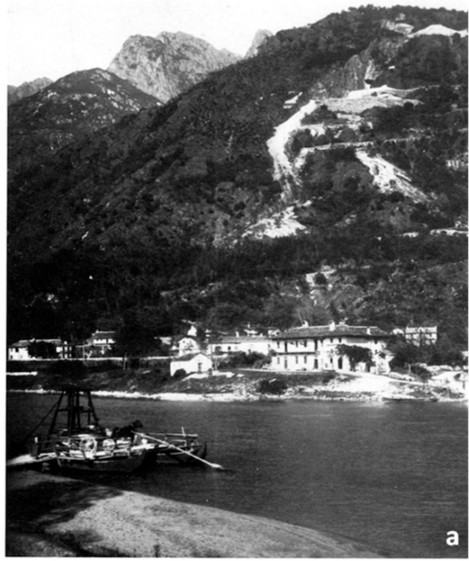
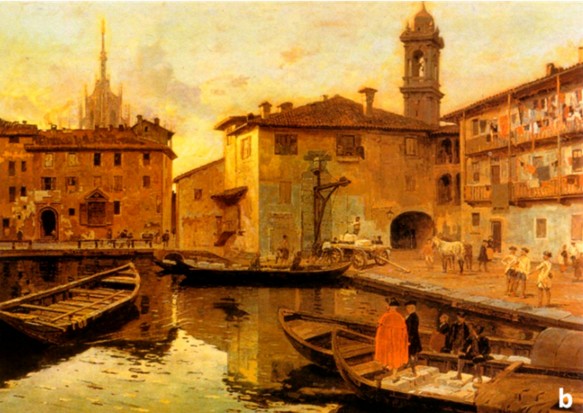

**Figure 10.** (**a**) Candoglia pier (late XIX–XX century). (**b**) Navigli di Milano during the Duomo di Milano construction activities (portrait). (Veneranda Fabrica Historical Archive).

Nowadays, all the quarried material is exploited and used in different applications: from the best quality (used to produce sculptures and masterpieces for the maintenance of the Cathedral) to irregular or not aesthetically suitable blocks (employed as armour stones).

The working activities are run out in Candoglia and Milan. In particular, the first block squaring by means of a diamond-wire block cutter is set in the working area close to the Cava Madre quarry.

The squared blocks are sent to the main working plant near Milan to be mechanically worked for the production of a first rough sketch of the sculpture. After this rough working phase, the blocks are sent back to the Candoglia mason stone cutter laboratory (near the offices of the Venaranda Fabrica del Duomo di Milano) to be chiselled by expert stone workers, to obtain the copy of the pieces to substitute (Figure 11).

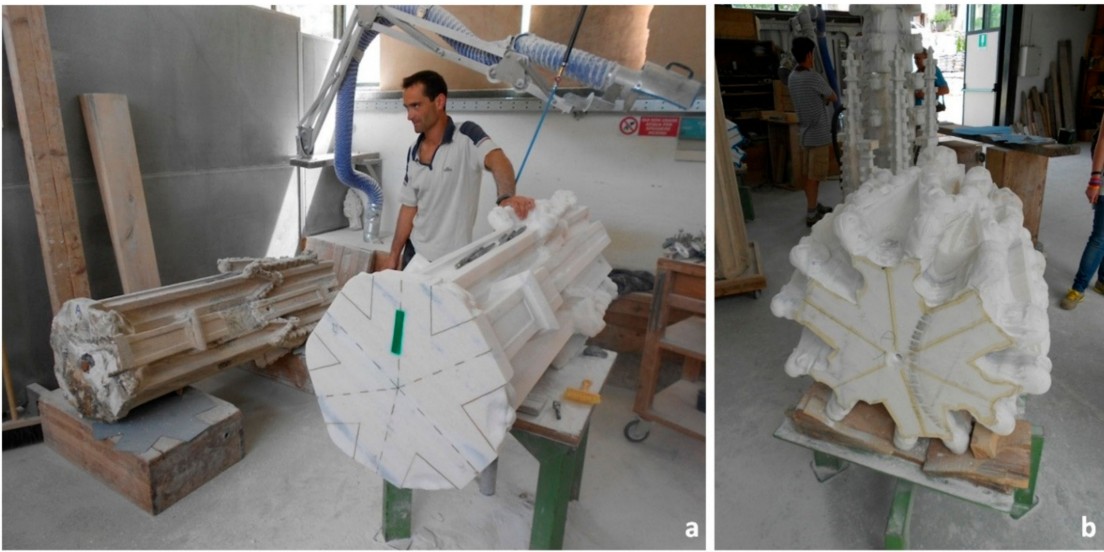

**Figure 11.** (**a**) Rough working sketch of the sculpture. (**b**) Chiselled sculpture to substitute the original one.

## 6. The Duomo di Milano and the Museum of the Venaranda Fabbrica del Duomo di Milano

The Milan Cathedral (Figure 12) is the seat of the Archbishop of Milan and is named to St Mary of the Nativity (Santa Maria Nascente). It could be considered as a "still alive" yard; indeed, Duomo di Milano has been completely built but its maintenance continues (the overlapping of maintenance with construction started in the mid-XIX century, when a large piece of the main spire fell onto the terraces below). The Cathedral is one of the most important and famous worldwide known Catholic church (5[th] place in term of size) and in Italy it comes after the Basilica of San Pietro, in Rome.

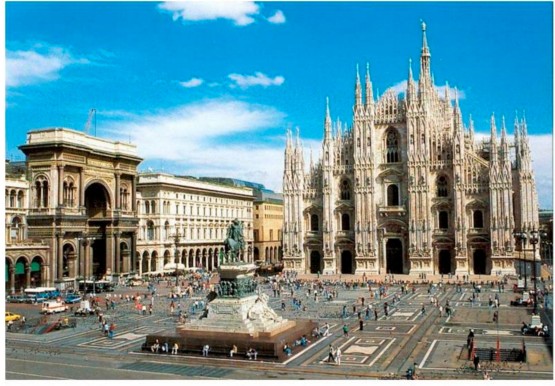
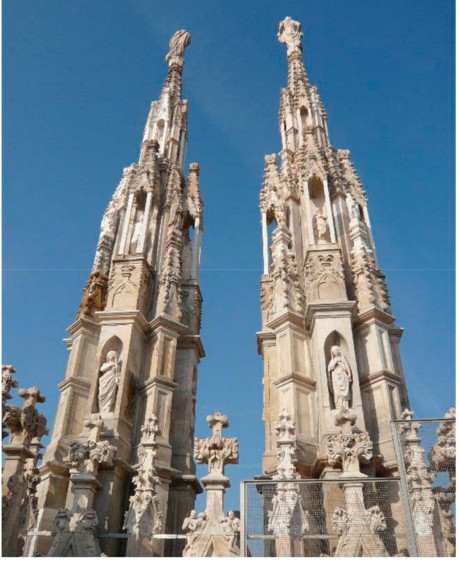

**Figure 12.** Duomo di Milano: general view from the main square and sketch of sculptures and steeples (different colours due to the different maintenance phases have to be highlighted).

The plan consists of a central nave with four side aisles, crossed by a trampset; opposite to the entrance it is possible to admire the choir and the apse. The main nave shows a 45 m height; it represents the he highest Gothic vaults of a completed church. Tourists can reach the roof, from which it is possible to appreciate some spectacular sculptures that otherwise would not be enjoyed.

The Museum of the Veneranda Fabbrica (Figure 13a) hosts several precious sculptures made of Candoglia marble, such as a Saint Ambrogio statue (Figure 13b).

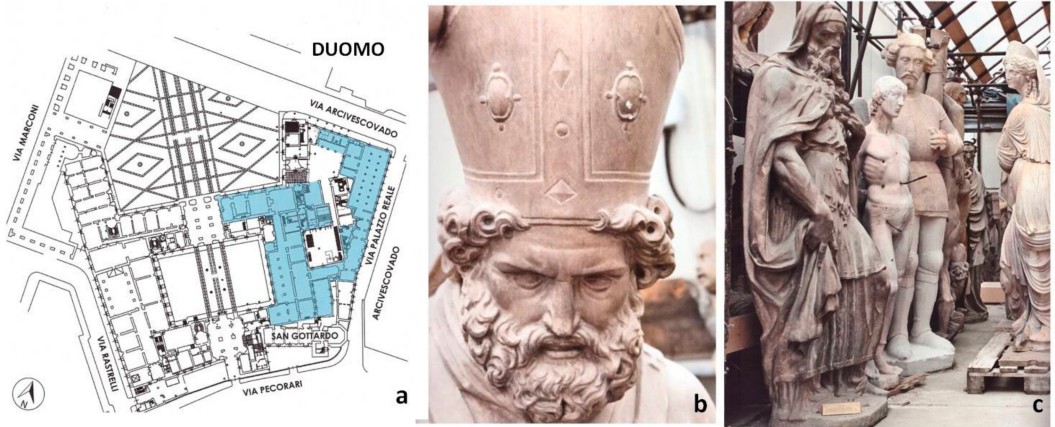

**Figure 13.** (**a**) Museum of the Veneranda Fabbrica del Duomo di Milano: located at Sforza palace, South of the Duomo. (**b**) Sant Ambrogio—Milan patron saint—sculpture. (**c**) sculptures stored in the museum.

## 7. Conclusions

Candoglia marble represents an example of heritage stone valorisation for the next generation: a unique and priceless material, a witness of the culture change of a wide territory in Piemonte and Lombardia (the quarrying area in Candoglia and the Milan cathedral). Indeed, thanks to its unique characteristics (physical, petrological and mineralogical) and to the peculiar history of its exploitation during the centuries for the construction and maintenance of an unique heritage building, Candoglia marble is worldwide well-known and appreciated.

Candoglia marble exploitation has influenced the culture and industrial development of an entire area over the centuries. Candoglia has to be deemed as a place where humankind and nature interact. Indeed, visiting the Candoglia area, the audience is fascinated by the immeasurable twist of natural landscape and human activities, which are tangible in:

- the need to discover and exploit a unique local resource, Candoglia marble, aesthetically suitable for the realisation and the maintenance of a worldwide known cultural and religious building(the Duomo di Milano);
- the presence of working and historical quarries which developed over the centuries from open pits to underground yard that reach into the heart of the mountain and exploit the most prospective part of the formation. Furthermore, the need to preserve the resource has brought about an upgrade of the quarrying technologies: from open pits exploited with explosives to underground pit yards, exploited with more modern techniques( the helicoidal and then the diamond wire machines);
- the continuing upgrade of working techniques, from local working activities (in Candoglia, Suna and Baveno) to the more recent activities managed both in Candoglia (in the quarry yard and the mason stone cutter laboratory) and Milan;
- the presence of historical means of transport, such as the "vie di lizza", the wagons being towed by pairs of oxen, the big barges, developed utilizing more modern transport by truck.

All these natural and human features of the area are fundamental to indicate the Candoglia marble area as geoheritage, with the consequence that Candoglia marble will be indicated for its designation as "Global Heritage Stone Resource".

A geotourist path "from quarry to building" can be organised, starting from the Candoglia quarry area (including both the still active Cava Madre and some yet accessible historic open pits) to the laboratory, and finally visiting the Duomo di Milano. It is also possible to consider a geotourist path "from building back to quarry", starting from the more famous Duomo di Milano, directing people to discover where everything started: an example of this approach is the path from the famous Arena of Arles back to the ancient quarries, exploited for the production of the employed stones.

**Author Contributions:** Conceptualization, G.A.D., A.B., D.C. and B.C.; Data curation, G.A.D., A.B., D.C., F.C. and E.C.; Funding acquisition, A.B. and D.C.; Investigation, G.A.D., A.B., D.C., F.C. and E.C.; Methodology, G.A.D., A.B. and B.C.; Supervision, G.A.D., A.B. and D.C.; Validation, G.A.D., D.C. and F.C.; Writing—original draft, G.A.D. and A.B.; Writing—review and editing, G.A.D., A.B., D.C. and F.C.

**Funding:** This research was funded by Compagnia San Paolo and University of Torino in the frame of the GeoDIVE Project, by MIUR (Italy) and CNR-IGG U.O of Torino (Italy).

**Acknowledgments:** Vittorio Barella and Enrico Allais (ISO4—Torino) are thanked for isotope analysis.

**Conflicts of Interest:** The authors declare no conflict of interest.

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
