# Peer review of "The Candoglia Marble and the “Veneranda Fabbrica del Duomo di Milano”: A Renowned Georesource to Be Potentially Designed as Global Heritage Stone"

_sustainability, doi:10.3390/su11174725_

Round 1
Reviewer 1 Report
The paper highlights the petrographic and technical characteristics of Candoglia marble, quarry extraction and processing , underlining the importance to assign to this ornamental stone the designation of Global Heritage Stone Resource.
Some check for the English and orthographic errors should be made.
Following some suggestions:
line 260: armourstone and not armour stone
caption of figure 2 (line 131): substitute pol. with polarized or cross nicols ,
heading of paragraph 3 and in the test, line 187: to substitute petrografic with petrographic,
substitute loc with region?
the terms “punciotto” in italian can be translate in English as plug and feather (Giornetti 1991, Stone Industry Technical Glossary): to verify in the manuscript and substitute,
line 222: 2nd and not 2o,
line 232: saint instead of sant.
Moreover some few integration text reviewer have been suggested.
To specify the percentage of calcite content for Candoglia marble in the paragraph 3
Table 1: please insert the m.u. in% also per other components
Table 2: simple compressive strength is uniaxial compressive strength? If yes please change. Please substitute freezing with freeze and thaw and specify the number of cycles if it possible. To check the values of indirect tensile strength: are the authors sure that the values refer to Brazilian test or the two values are of Brazilian test and of flexural strength: they are too much different.
Change the heading of paragraph 5 in Candoglia marble exploitation and processing, to best refer to the content of paragraph.
line 203: please to insert to reference
Author Response
Dear Reviewer,
We corrected the paper accordingly to your revisions (together with the ones suggested by reviewers #2 and #3, and specific references have been added.
English has been checked by a native english speker (one of the author of the paper).
In particular:
ine 260: armourstone and not armour stone: DONE
caption of figure 2 (line 131): substitute pol. with polarized or cross nicols : DONE
heading of paragraph 3 and in the test, line 187: to substitute petrografic with petrographic: DONE
substitute loc with region?: SUBSTITUTED
the terms “punciotto” in italian can be translate in English as plug and feather (Giornetti 1991, Stone Industry Technical Glossary): to verify in the manuscript and substitute: DONE
line 222: 2nd and not 2o, DONE
line 232: saint instead of sant. DONE
Moreover some few integration text reviewer have been suggested.
To specify the percentage of calcite content for Candoglia marble in the paragraph 3: 80-85% CaCO3 and 15-20% other minerals (as now reported in the text)
Table 1: please insert the m.u. in% also per other components: DONE
Table 2: simple compressive strength is uniaxial compressive strength? If yes please change. Please substitute freezing with freeze and thaw and specify the number of cycles if it possible. To check the values of indirect tensile strength: are the authors sure that the values refer to Brazilian test or the two values are of Brazilian test and of flexural strength: they are too much different.: CHANGED. As for the indirect tensile stregth, the vaues reported come from bibliographic data.
Change the heading of paragraph 5 in Candoglia marble exploitation and processing, to best refer to the content of paragraph. DONE
line 203: please to insert to reference DONE
thank you for the effort put in revision phase and for the nice suggestions you gave us.
best wishes
Giovanna DINO
on the behalf of the authors.

Reviewer 2 Report
General Comments
The manuscript titled “Candoglia Marble and the "Veneranda Fabbrica del Duomo di Milano": a resource for Global Heritage Stone Designation in the Italian Alps” by Giovanna Antonella Dino and co-authors, aims at explain the use of the Candoglia Marble as resource and how its use has influenced the culture development over the centuries.
However, there are some unclear points, for example when the authors describe the minero-petrographic and geochemical results. In detail, it is unclear whether the authors performed analyses on samples, or whether they used only the bibliography. In the first case, if they have used samples, they must add paragraphs on "Sampling” and “Analytical techniques". If, on the other hand, they only referred to the bibliography, they will have to make this point clearer. Otherwise it is not clear when they talk about results on samples (see table 1 for example), which shows minero-petrographic and geochemical data.
Instead, the paragraphs on "Candoglia marble" exploitation are clear and well developed.
Therefore, in general, the authors must better connect the two parts, the one more focused on “a scientific approach” and the other related to the “exploitation and use of the material”.
In conclusion, the approach well fits the proposed goals; nevertheless the paper suffers of some shortcomings that must be fixed before publication on Sustainability.
Authors are suggested to deal with the following recommendations:
Abstract
Line 17
On the first line, after "for several famous applications" it would be necessary that the authors stress the use of these marbles as construction materials (or uses related to cultural heritage, such as statuary, sculptures, etc.). Therefore I suggest reformulating the sentence and making clear this use of marble as one of the most important building material since the past.
Introduction
Line 46
Please change “archeometric” in “archaeometric”
Line 47
It is needful to add proper references, there are so many works on this topic. Please add: 1) Attanasio D., Brilli M., Ogle N. 2006. The Isotopic Signature Of Classical Marbles. 2) Gorgoni C., Lazzarini L., Pallante P. & Turi B. 2002. An Updated And Detailed Mineropetrographic And C-O Stable Isotopic Reference Database For The Main Mediterranean Marbles Used In Antiquity. 3) Moens L., Roos P., De Rudder J., De Paepe P., Van Hende J., Waelkens M. 1988. A Multi-Method Approach To The Identification Of White Marbles Used In Antique Artifacts. 4) Belfiore C M, Ricca M , La Russa M F , Ruffolo S A , Galli G , Barca D , Malagodi M , Vallefuoco M , Sprovieri M , Pezzino, Provenance study of building and statuary marbles from the Roman archaeological site of “Villa dei Quintili (Rome, Italy). Italian Journal of Geosciences, 2016.
Line 75
Please specify on the "Candoglia Marble" because on other marbles these archaeometric techniques have been widely used.
Geological Setting
In figure 1, the authors describe figures 1a and 1b. They should indicate the letters A and B in the figure itself, and not only in the caption.
Doubts after the paragraph “Geological Setting”
After the "Geological Setting" the authors add a section "Petrografic Characterisation". The connection between these two paragraphs is unclear. Has it been conducted a sampling? Where? On samples from quarry? Please explain better and eventually add a "Sampling" paragraph. In fact in Table 1 the authors insert the properties of 6 samples.
Add also info about the analytical techniques used (optical polarizing microscope, XRF, etc.). The results have been described (in the following paragraphs) but not the techniques. Please add a paragraph with the techniques used in this study.
Petrografic Characterisation
In the "Petrografic Characterisation" section, the authors also describe the geochemical properties (isotopes, minor and trace element, etc.). Therefore the name of the paragraph should be "Petrografic and geochemical Characterization" or alternatively, the author should separate the two paragraphs.
Figure 2 The same comment of figure 1 (add A and B in the figure). Also, in the caption, what do the authors mean for "Optical microscope, only pol."? The use of only one Nicol? Perhaps, the observation at crossed Nicols?
Line 146
Please change “archeometric” in “archaeometric”
Please change “minero-petro” in “minero-petrographic”
Please change “to have updated info about its physico-mechanical behavior” in “to know its physico-mechanical behavior”.
Physico-mechanical characterization
Where the physico-mechanical behaviors have been tested? On which samples? Even in this section, as in the previous paragraphs, it is unclear where these measures were performed.
As for the tests (water absorption, compressive strength after freezing and impact strength) the authors must be clear and add the reference legislation followed to conduct the tests and/or the bibliography.
Candoglia marble exploitation
In all the figures authors should indicate the letters A and B, and not only in the caption. Only figure 13 shows this detail (A, B, C)
Conclusion
Please reformulate the sentence “it is a gift for people, who have to protect and respect both the original ornamental stone and the place interested by the presence of this unique and priceless material”
Author Response
Dear Reviewer
we've chaked and changed the paper accordingly to your comments and suggestions.
English has been checked by a native english speaker (one of the author) and references have been added.
In particular:
Abstract
Line 17
On the first line, after "for several famous applications" it would be necessary that the authors stress the use of these marbles as construction materials (or uses related to cultural heritage, such as statuary, sculptures, etc.). Therefore I suggest reformulating the sentence and making clear this use of marble as one of the most important building material since the past: DONE
Introduction
Line 46
Please change “archeometric” in “archaeometric”: DONE
Line 47
It is needful to add proper references, there are so many works on this topic. Please add: 1) Attanasio D., Brilli M., Ogle N. 2006. The Isotopic Signature Of Classical Marbles. 2) Gorgoni C., Lazzarini L., Pallante P. & Turi B. 2002. An Updated And Detailed Mineropetrographic And C-O Stable Isotopic Reference Database For The Main Mediterranean Marbles Used In Antiquity. 3) Moens L., Roos P., De Rudder J., De Paepe P., Van Hende J., Waelkens M. 1988. A Multi-Method Approach To The Identification Of White Marbles Used In Antique Artifacts. 4) Belfiore C M, Ricca M , La Russa M F , Ruffolo S A , Galli G , Barca D , Malagodi M , Vallefuoco M , Sprovieri M , Pezzino, Provenance study of building and statuary marbles from the Roman archaeological site of “Villa dei Quintili (Rome, Italy). Italian Journal of Geosciences, 2016.
DONE
Line 75
Please specify on the "Candoglia Marble" because on other marbles these archaeometric techniques have been widely used. The sentence has been updated (line 76)
Geological Setting
In figure 1, the authors describe figures 1a and 1b. They should indicate the letters A and B in the figure itself, and not only in the caption. DONE
Doubts after the paragraph “Geological Setting”
After the "Geological Setting" the authors add a section "Petrografic Characterisation". The connection between these two paragraphs is unclear. Has it been conducted a sampling? Where? On samples from quarry? Please explain better and eventually add a "Sampling" paragraph. In fact in Table 1 the authors insert the properties of 6 samples.
Add also info about the analytical techniques used (optical polarizing microscope, XRF, etc.). The results have been described (in the following paragraphs) but not the techniques. Please add a paragraph with the techniques used in this study.
A chapter about "characterisation: materials and methods" has been added; info about sampled materials and bibliographic data have been inserted
Petrografic Characterisation
In the "Petrografic Characterisation" section, the authors also describe the geochemical properties (isotopes, minor and trace element, etc.). Therefore the name of the paragraph should be "Petrografic and geochemical Characterization" or alternatively, the author should separate the two paragraphs. DONE
Figure 2 The same comment of figure 1 (add A and B in the figure). Also, in the caption, what do the authors mean for "Optical microscope, only pol."? The use of only one Nicol? Perhaps, the observation at crossed Nicols? DONE
Line 146
Please change “archeometric” in “archaeometric” DONE
Please change “minero-petro” in “minero-petrographic” DONE
Please change “to have updated info about its physico-mechanical behavior” in “to know its physico-mechanical behavior”. DONE
Physico-mechanical characterization
Where the physico-mechanical behaviors have been tested? On which samples? Even in this section, as in the previous paragraphs, it is unclear where these measures were performed. Data from bibliography: a chapter about charatcerisation M&M has been inserted, reporting that data have been collected from published papers and documents
As for the tests (water absorption, compressive strength after freezing and impact strength) the authors must be clear and add the reference legislation followed to conduct the tests and/or the bibliography. DONE
Candoglia marble exploitation
In all the figures authors should indicate the letters A and B, and not only in the caption. Only figure 13 shows this detail (A, B, C). DONE
Conclusion
Please reformulate the sentence “it is a gift for people, who have to protect and respect both the original ornamental stone and the place interested by the presence of this unique and priceless material”.
the above mentioned sentence has been changed (lines 279-282)
Thank you for the effort puts in the revision phase
best regards
Giovanna DINO, on the behalf of the authors.

Reviewer 3 Report
The manuscript under review deals with one of the well known ornamental stones, the Candoglia Marble, which became famous because of its use in the “Duomo di Milano”. The authors are following a very interesting multi-analytical approach for first characterizing the minero-petrographic and physical properties and then describe the human activities connected to this earth resource, which make it a good example of the Global Heritage Stone Resource. The manuscript however needs to be further improved in certain aspects as is indicated bellow.
General points
· A section about the methods employed and the materias analyzed should be added.
· Does “Section 4” contain results of the present research or from the literature? Authors should made clear their own results.
· Sections 5 and 6 are missing the relative references, mainly for the photographs used in the images
Specific points
Page 1, Line 38: change “…from Greek and Roman ages” to “…from Greek and Roman times”
Page 2, Line 46: change “archeometric” to “archaeometric”
Page 2, Line 47: give some examples for references about provenance databases of Classical Marbles
Page 2, Line 48: change “Generally, the uses…” to “Generally, the stone uses…”
Page 2, Line 63: change “…to widespread…” to “…to disseminate…”
Page 3, Line 112: change “Petrografic Characterisation” to “Petrographic Characterisation”
Page 3, Lines 120-122: Add references for the Mediterranean marbles you are reffering to
Page 4, Lines 123-126: English language should be improved
Page 3, Line 129: change “b. microphotograph of marble, showing heteroblastic grain,…” to “b. photomicrograph of marble, showing heteroblastic texture,…”
Page 3, Line 131: change “Optical microscope, only pol.” to “Polarizing microscope, crossed polars”
Page 4, Table 1: The three last parameters are the same for every sample analysed. They could be omitted and only be mentioned in the caption of the table.
Page 5, Lines 155-157: The English language in the last sentence should be improved.
Page 10, Line 222: change “5o” to “5th” and “2o” to “2nd”.
It is the reviewer belief that the manuscript should be first improved taking into consideration the points highlighted above and then could be accepted for publication.
Author Response
Dear Reviewer
we corrected the paper accordingly to your comments and suggestions, together with the ones by reviewer #1 and #2.
English check by a native english speaker (one of the author)
references have been added.
In detail, answering your specific comments:
General points
· A section about the methods employed and the materias analyzed should be added. DONE
· Does “Section 4” contain results of the present research or from the literature? Authors should made clear their own results. LITERATURE DATA, explained better in the text
· Sections 5 and 6 are missing the relative references, mainly for the photographs used in the images REFERENCES HAVE BEEN ADDED
Specific points
Page 1, Line 38: change “…from Greek and Roman ages” to “…from Greek and Roman times” DONE
Page 2, Line 46: change “archeometric” to “archaeometric” DONE
Page 2, Line 47: give some examples for references about provenance databases of Classical Marbles DONE
Page 2, Line 48: change “Generally, the uses…” to “Generally, the stone uses…” DONE
Page 2, Line 63: change “…to widespread…” to “…to disseminate…” DONE
Page 3, Line 112: change “Petrografic Characterisation” to “Petrographic Characterisation” DONE
Page 3, Lines 120-122: Add references for the Mediterranean marbles you are reffering to DONE
Page 4, Lines 123-126: English language should be improved DONE
Page 3, Line 129: change “b. microphotograph of marble, showing heteroblastic grain,…” to “b. photomicrograph of marble, showing heteroblastic texture,…” DONE
Page 3, Line 131: change “Optical microscope, only pol.” to “Polarizing microscope, crossed polars” DONE
Page 4, Table 1: The three last parameters are the same for every sample analysed. They could be omitted and only be mentioned in the caption of the table. we decide to insert the info in tab. 1, even if they report the same values, because we think it is important to highligh the specific achaeometric parameters.
Page 5, Lines 155-157: The English language in the last sentence should be improved. DONE
Page 10, Line 222: change “5o” to “5th” and “2o” to “2nd”. DONE
Thank you for your effort in reviewing our paper
best regards
Giovanna DINO, on behalf of the authors
